# Comparative Transcriptome Analysis of the Expression of Antioxidant and Immunity Genes in the Spleen of a Cyanidin 3-O-Glucoside-Treated Alzheimer’s Mouse Model

**DOI:** 10.3390/antiox10091435

**Published:** 2021-09-09

**Authors:** Varun Jaiswal, Miey Park, Hae-Jeung Lee

**Affiliations:** 1Department of Food and Nutrition, College of BioNano Technology, Gachon University, Seongnam-si 13120, Gyeonggi-do, Korea; computationalvarun@gmail.com (V.J.); mpark@gachon.ac.kr (M.P.); 2Institute for Aging and Clinical Nutrition Research, Gachon University, Seongnam-si 13120, Gyeonggi-do, Korea

**Keywords:** cyanidin 3-O-glucoside, Alzheimer’s disease, transcriptome, antioxidant, immune modulation, genes, mechanisms

## Abstract

Cyanidin 3-O-glucoside (C3G) is a well-known antioxidant found as a dietary anthocyanin in different fruits and vegetables. It has protective and therapeutic effects on various diseases. It can reduce neuronal death from amyloid-beta (Aβ)-induced toxicity and promote the inhibition of Aβ fibrillization. Antioxidant and immune modulation might play a critical role in the properties of C3G against Alzheimer’s disease (AD) and other diseases. However, limited studies have been performed on the mechanism involved in the effect of C3G through transcriptome analysis. Thus, the objective of this study was to perform comparative transcriptome analysis of the spleen to determine gene expression profiles of wild-type mice (C57BL/6J Jms), an Alzheimer’s mouse model (APPswe/PS1dE9 mice), and a C3G-treated Alzheimer’s mouse model. Differentially expressed antioxidant, immune-related, and AD pathways genes were identified in the treated group. The validation of gene expression data via RT-PCR studies further supported the current findings. Six important antioxidant genes (S100a8, S100a9, Prdx2, Hp, Mpst, and Prxl2a) and a high number of immune-related genes were found to be upregulated in the treatment groups, suggesting the possible antioxidant and immunomodulatory mechanisms of C3G, respectively. Further studies are strongly recommended to elucidate the precise role of these essential genes and optimize the therapeutic function of C3G in AD and other disease conditions.

## 1. Introduction

Cyanidin 3-O-glucoside (C3G) is an important dietary anthocyanin that is present in different fruits, vegetables, and grains. Studies are increasingly reporting that C3G has protective effects and therapeutic potential against diseases such as diabetes [1,2,3], obesity [4], cardiovascular disease [2,5], neurological diseases [6], asthma [7], atherosclerosis [8], Alzheimer’s disease (AD) [6], and different types of cancer [9,10,11,12,13]. These multiple pharmacological properties of C3G might be primarily attributed to its antioxidant and immunomodulatory effects [10,14,15,16]. In the past, a few studies have investigated genes and elucidated the mechanism involved in the immunomodulation and antioxidant properties of C3G. Therefore, the current study was designed to investigate the effect of C3G on the whole-transcriptome expression of the spleen to discover important immune-modulating and antioxidant system genes using wild-type and AD mouse models.

In Korea, AD was reported to be the ninth leading cause of death [17], and deaths caused by AD are expected to increase [17]. AD is the sixth leading cause of death in the United States [18], with an estimated more than six million individuals diagnosed with AD in the age group of 65 years or above in 2021 [19]. The occurrence of AD in 2050 is expected to be double the current level [19]. C3G is protective against glutamate-induced neuronal cell death [20] and ischemia-induced neuronal cell death [21,22]. It can prevent ethanol neurotoxicity [23] and can protect against amyloid β(Aβ)-induced cytotoxicity [23,24,25]. It can also promote the inhibition of Aβ (Aβ40, Aβ1-42) fibrillization and protect neuronal cells from Aβ1-42, Aβ40, and Aβ25-35 induced cytotoxicity [6,24,25,26,27]. In rats, C3G can prevent the cognitive impairment induced by Aβ [28]. The in vitro and in vivo protection of C3G against neuronal cell death (other than Aβ-induced cell death) suggests its protective effect via other mechanisms in addition to the inhibition of Aβ fibrillization [20,21,22,23]. Although antioxidant activity and immune modulation were observed following C3G administration in different studies, the molecular genetic mechanisms underlying its immune modulation in AD pathogenesis are not yet well-studied [29]. The spleen is one of the most important organs in the immune system. It mediates a wide range of immunological functions [30]. The spleen has also been used to study antioxidant activity in different organisms [31,32,33,34]; therefore, we conducted gene expression studies with spleens obtained from a mouse model of AD following C3G treatment to identify the important genes and mechanisms for immune modulation, antioxidant activity, and subsequent AD pathogenesis. Whole-transcriptome analysis via RNA-Seq data combined with computational analysis can provide insights into different biological mechanisms [35,36,37,38,39,40,41]. In this study, analysis of the entire transcriptome of the C3G-treated mouse model of AD compared with the untreated mouse model of AD and wild-type mice was used to identify differentially expressed genes (DEGs) and transcripts. These DEGs, involved in the clearance of reactive oxygen species, inflammation, immune response, and innate immunity, are expected to provide insight into the antioxidant and immunomodulatory mechanisms of C3G and its possible role in AD and other diseases [38].

## 2. Materials and Methods

### 2.1. Materials and Animal Model

The C3G (cyanidin-3-O-glucoside (C_21_H_21_O_11_, Cat. No. CFN99740)) used in the study was purchased from ChemFaces (Wuhan, Hubei, China). Animals used in this study included wild-type mice (C57BL/6J Jms) purchased from Hamamatsu-shi, Shizuoka, Japan, grouped as Wt mice, and AD model mice (APPswe/PS1dE9 mice) purchased from the Jackson laboratory [42], designated as ADM mice. AD model mice treated with C3G were grouped as ADM mice+C3G. All animals used in the study were nine months old. They underwent two weeks of acclimatization before starting the experiment. Mice (3 mice in each group) were randomly selected without any bias. The C3G dose administered by oral gavage for the ADM mice+C3G group was 30 mg/kg/day for 38 weeks. These mice were provided ad libitum access to food and water during this study. The dose and study period were decided according to similar studies that investigated the effect of C3G [1,9,27,43]. This animal study was conducted in accordance with guidelines stipulated by the Ministry of Food and Drug Safety for the Care and Use of Laboratory Animals. It was approved by the Institutional Animal Care and Use Committee of Eulji University School of Medicine (EUIACUC 20-13). All efforts were made to minimize the number and suffering of animals used in the study. Mice were anesthetized using CO_2_. Their spleens were removed for RNA extraction.

### 2.2. RNA-Seq Library Preparation and Sequencing

Spleen tissue samples were homogenized to extract total RNA using a TRIzol reagent (Thermo Fisher Scientific, Waltham, MA, USA)-based method. Total RNA (500 ng) was used to prepare the entire transcriptome sequencing library. The whole-transcriptome RNA was enriched by depleting ribosomal RNA (rRNA) to generate the whole-transcriptome sequencing library using MGIEasy RNA Directional Library Prep Kit (MGI) according to the manufacturer’s instructions. The remaining RNA was fragmented at an elevated temperature. These cleaved RNA fragments were used to synthesize first-strand cDNA with reverse transcriptase and random primers. Strand specificity was achieved with RT directional buffer, followed by second-strand cDNA synthesis. These cDNA fragments contained an additional single A base and subsequent ligation of the adapter. Products were then purified and enriched with PCR to create the final cDNA library. The double-stranded cDNA library was quantified using a QauntiFluor ONE dsDNA System (Promega, Madison, WI, USA). It had 330 ng in a total volume of 60 mL or less. The library was incubated at 37 °C for 60 min and then digested at 37 °C for 30 min, followed by a cleanup of circularization product. The library was incubated at 30 °C for 25 min using DNB enzyme to obtain a DNA nanoball (DNB). Finally, the library was quantified using a QauntiFluor ssDNA System (Promega). The prepared DNB was sequenced using an MGIseq system (MGI) with 150 bp paired-end reads.

### 2.3. RNA-Seq Analysis of Assembly and Differential Expression of Genes

Paired-end reads from all 9 samples were used for RNA-Seq data analysis of three different groups (NM, AM, and CAM) to assemble the whole transcriptome for comparative analysis. Quality control (QC) is important when initially selecting good-quality reads for further processing. AfterQC was used for QC, which entailed filtering, trimming, and error removal [44]. Good-quality reads were obtained as the output from the QC step for all 9 samples. They were used for the alignment with the reference genome (GRCm38, the mouse reference genome assembly released by the Genome Reference Consortium) using HISAT2 [45]. Alignment files in Sequence Alignment Map (SAM) format were converted to BAM files through SAMtools [46] for further assembly. Finally, StringTie assembler [47] was used to assemble alignment files (BAM format). The assembler was used with the -e option, which required a combination of the output of all samples for further differential expression analysis using EdgeR [48] and DESeq2 [49]. Default statistical cutoff parameters for identifying DEGs (FDR < 0.1 and minimum 2-fold change) were used in DESeq2.

### 2.4. Function Enrichment of DEGs

Differentially expressed genes (DEGs) were subjected to functional enrichment analysis using protein analysis through evolutionary relationship (PANTHER) [50] and AllEnricher [51]. PANTHER was used to perform enrichment analysis based on biological process, cellular component, molecular function, protein class, and pathway enrichment [50]. AllEnricher was used for functional interpretation based on gene ontology (GO) [52], Reactome [53], and visualization through a bubble plot. The latest libraries of these resources were downloaded and used for the analysis.

### 2.5. Comparison of DEGs and Identification of Antioxidant and Immune Genes

Comparison of the DEGs identified in differential expression analysis between Wt mice vs. ADM mice and ADM mice+C3G vs. ADM mice was performed using InteractiVenn [54] to identify the important genes that were up- or downregulated. DEGs from the comparison between ADM mice and ADM mice+C3G were used to identify antioxidant genes through the antioxidant protein database (AOD), which contains antioxidant proteins reported in the literature [55]. All antioxidant proteins reported in mice were collected from the AOD database and converted to gene ID with the help of g:Profiler [56]. Venn analysis was used again to identify the DEGs present in the collected gene IDs. The DEGs identified in the comparison between ADM mice and ADM mice+C3G were also compared with immune-related gene sets. Two gene sets (immune gene set 1 and immune genes set 2) were prepared for the comparison. Immune gene set-1 (ImmGS1) is composed of all immunity-related genes derived from the immunome database, which collects genes known to be directly involved in immune mechanisms reported in the literature [57]. Similarly, immune gene set 2 (ImmGS2) was prepared from genes reported in innateDB, which collects genes involved in innate immunity [57]. As genes in innateDB were obtained from human,; homologous genes were searched for in mice using g:Profiler [56]. Finally, all genes associated with innate immunity in mice were converted to gene accession numbers to create ImmGS2. Both gene sets (ImmGS1 and ImmGS2) were used for comparison with the DEGs to identify common genes in different comparisons via set analysis. InteractiVenn was used to perform set analysis to create a Venn diagram and prepare the list of common genes [54].

### 2.6. Protein–Protein Interaction Network

All DEGs from both comparisons were used to construct a protein-protein interaction network (PPIN) using STRING-version11.0 [58]. Lists of the DEGs from both comparisons (Wt mice vs. ADM mice and ADM mice+C3G vs. ADM mice) were used separately to construct two different PPINs. PPIN and KEGG network enrichment analyses were conducted using *Mus musculus* as the target organism in the server. The default settings were used in this study. Statistical parameters such as *p*-value for analysis were calculated and the results were extracted in tab-separated variables and PNG image formats.

### 2.7. Quantitative Real-Time PCR (qRT-PCR) Assay to Validate the Expression of Important Genes

DEGs that were up- or downregulated in both comparisons, Wt mice vs. ADM mice and ADM mice+C3G vs. ADM mice (except the genes not expressed in samples belonging to more than one group), were selected for validation via RT-PCR. The gene sequences of all selected genes were obtained from Ensemble [59]. Primers were designed using the primer3web server. Primer sequences are provided in Appendix A.

## 3. Results

### 3.1. Quality Control and Alignment of RNA-Seq Reads

A total of 212,745,311 reads were obtained from the RNA-Seq analysis of all nine samples, with an average of 23,638,367.8 reads per sample (Table 1). Preprocessing is the initial step of RNA-Seq analysis, mainly used for QC. A high percentage (>95%) of good reads was obtained for all the samples, with an average of 96.4% (Table 1). Subsequently, a high percentage (>91%) of overall alignment with the mouse reference genome (GRCm38) was achieved, with an average of 92.9% for all the samples (Table 1).

### 3.2. Assembly and Differential Expression Analysis

The whole transcriptome was built via the assembled reads obtained from the alignment files. The expression was determined in the form of fragments per kilobase of transcript per million mapped reads (FPKM). Read count and FPKM values were calculated for each assembled gene/transcript. Furthermore, the gene count of all nine samples was used in the differential-expression-related analyses as the preferred input. The preprocessed results are graphically presented as a read count bar plot, a distribution of transformed data plot, and a density plot of transformed data (Appendix A). The first and second principal components were used for the PCA plot, which shows the difference between the groups in this study. The samples from the control group cluster in the right portion of the graph. The samples from the ADM group and the ADM +C3G groups cluster on the opposite side. The samples from the ADM group cluster above the ADM+C3G group (Appendix A). Similar graphs were obtained from multidimensional scaling and t-SNE options [60] (Appendix A). The differential expression analysis via DEseq2 revealed up- and downregulated genes among the different groups used in the study [61]. A comparison of the ADM mice with the Wt mice revealed the upregulation of 444 genes and the downregulation of 904 genes. The comparison of the ADM mice and the ADM mice+C3G revealed 487 upregulated genes and 53 downregulated genes (Appendix A). The heat map, MA, and volcano plots of DEGs were derived from both the comparisons (Figure 1 and Figure 2). The selected DEGs were further enriched and analyzed.

### 3.3. Functional Enrichment Analysis of DEGs

Enrichment analysis was performed according to the biological process, cellular component, molecular function, protein class, pathway, and reactome pathway analysis. The upregulated DEGs in the Wt mice, compared with those in the ADM mice, were enriched in 18 biological processes, 3 cellular components, 7 molecular functions, 18 protein classes, 31 pathways, and 337 reactome pathways (Appendix A). The downregulated DEGs in the Wt mice, compared with the ADM mice, were enriched in 19 biological processes, 3 cellular components, 8 molecular functions, 22 protein classes, 80 pathways, and 334 reactome pathways (Appendix A). Similarly, the upregulated DEGs in the ADM mice+C3G group, compared with the ADM mice, were enriched in 16 biological processes, 3 cellular components, 8 molecular functions, 20 protein classes, 55 pathways, and 116 reactome pathways (Appendix A). The downregulated DEGs in the ADM mice+C3G group, compared with the ADM mice, were enriched in 10 biological processes, 3 cellular components, 4 molecular functions, 10 protein classes, and 9 pathways (Appendix A).

### 3.4. Identification of Common DEGs in Comparison

Five genes (Slpi, Oas2, Gm15133, Ighv11-2, and Nnt) were found to be upregulated in both the Wt mice vs. ADM mice and the ADM mice+C3G vs. ADM mice comparisons (Table 2). Similarly, four genes (Cd209e, D630045J12Rik, Gm10260, and Igkv8-28) were found to be downregulated in both the Wt mice vs. ADM mice and the ADM mice+C3G vs. ADM mice comparisons. However, 253 and 6 DEGs showed opposite expression patterns, i.e., 253 DEGs were downregulated in the Wt mice vs. ADM mice and upregulated in the ADM mice+C3G vs. ADM mice comparisons, respectively, and vice versa (Figure 3A).

### 3.5. Identification of DEGs with Antioxidant Activity

Comparing the DEGs between the ADM mice+C3G and the ADM mice and the known antioxidant genes resulted in the identification of six antioxidant genes (S100a8, S100a9, Prdx2, Hp, Mpst, and Prxl2a), which were upregulated in the C3G treatment group (Table 2). No antioxidant gene was downregulated in the ADM mice+C3G group (Figure 3B).

### 3.6. Identification of DEGs with Immune-Related Function

When the DEGs were analyzed for ImmGS1 (comprising the immune genes directly involved in immune-related processes), 28 genes and 1 gene were found to be upregulated and downregulated, respectively, in the ADM mice+C3G than in the ADM mice. When the DEGs were analyzed for ImmGS2 (innate immunity genes), 33 were upregulated and 1 gene was downregulated in the ADM mice+C3G than in the ADM mice (Figure 3C).

### 3.7. PPIN Analysis

Two separate protein-protein interaction networks, PPIN1 and PPIN2, were drawn from the DEGs identified in the ADM mice vs. Wt mice and the ADM mice+C3G vs. ADM mice comparisons, respectively (Appendix A). A highly connected network was observed in both analyses. In PPN1, the total number of nodes was 839, with 3748 edges and an average node degree of 8.93, although the expected number of edges was only 1986 (Appendix A). Similarly, in PPIN2, a total of 6429 edges were observed between 463 nodes, with an average node degree of 27.8, although the expected number of edges was 1364 (Appendix A). The PPI enrichment *p*-values were less than 1.0 × 10^−16^ in both the PPIN1 and PPIN2 networks (Appendix A).

### 3.8. qRT-PCR Assay

The expressions of selected genes that were up- or downregulated in both comparisons, i.e., Wt mice vs. ADM mice and ADM mice+C3G vs. ADM mice in the RNA-Seq analysis, were validated using all nine samples via qRT-PCR assays. All the genes followed a similar expression pattern in each group (up- or downregulated in both comparisons) according to the RNA-Seq analysis (Figure 4).

## 4. Discussion

Comparative whole-transcriptome analysis of C3G-treated ADM mice was conducted to identify the genes and subsequently infer possible molecular mechanisms underlying the antioxidant and immunomodulatory properties of C3G observed in a number of studies [7,10,14,15,20,27]. Further analysis considering the importance of these properties in various diseases was conducted to explore the genes and the potential mechanism involved in the antioxidant and immune-modulation effects of C3G to develop it as a possible therapeutic not only for AD, but also for other disease conditions. In an earlier study, C3G also alleviated the cognitive impairment in the same Alzheimer’s mouse model [27]. Neuroinflammation in the brain might be associated with the immune response in the whole body [62]. Therefore, we wanted to focus on the antioxidant and immunomodulatory activities of C3G in the spleen of the APPswe/PS1dE9 mouse model of AD [63]. Spleen tissue has been used to investigate the immunomodulatory and antioxidant effects of phytochemicals in animal models [33,34]. The transcriptome of the spleen, one of the important organs of the immune system, was generated and comparatively analyzed among the Wt mice, ADM mice, and ADM mice+C3G (treated with C3G) groups. The transcriptomic sequences obtained from the samples derived from the different groups were assembled using a reference-based assembly method, the method of choice for an organism with a high-quality genome sequence available. The high percentage (>95%) of good-quality reads (Table 1) and the high alignment rate (>92%) of reads with the reference genome in all nine samples verified the high quality of the samples used in the study (Table 1). The unique transcriptomic resource of C3G-treated mice generated in the current study can also be used in future studies. Standard pipelines [47,61] were used for the identification of gene expression and differential gene expression analysis in all the groups. Pathway enrichment analysis revealed that the pathways associated with the immune response such as inflammation-mediated chemokines and cytokines, the integrin pathway, gamma-aminobutyric acid (GABA) B receptor signaling, FAS signaling, and antioxidant activity such as the oxidative stress response (P00046) were enriched in DEGs upregulated in the ADM mice+C3G group (Appendix A). The genes that were up- or downregulated in both comparisons might be important in AD pathogenesis, with the ADM mice+C3G group as the treatment group compared with AM; the Wt mice represent a group of wild-type mice. Thus, the differences between both the Wt mice and the CADM mice compared with the ADM mice might be associated with AD pathology. Five genes (Slpi, Oas2, Gm15133, Ighv11-2, and Nnt) were found to be upregulated in both comparisons. Four (Slpi, Oas2, Ighv11-2, and Nnt) of these five genes are associated with the immune response [64,65,66,67]. The one remaining gene (Gm15133) has not been well-studied and is listed as a predicted gene. The involvement of most of these upregulated genes in the immune response/immune modulation and the association of two (Slpi and Nnt) of these genes with AD suggests the need to investigate all of these five genes in the future as candidate genes underlying the immunomodulatory mechanism in AD [68,69]. Interestingly, Slpi and Nnt are known to contribute to antioxidant properties, as reported in the literature [70,71]. Similarly, four genes (Cd209e, D630045J12Rik, Gm10260, and Igkv8-28) were found to be downregulated in both comparative analyses. Of them, two (Cd209e and Igkv8-28) might be associated with the immune response [72,73]. The analysis of these important DEGs in both comparisons via RT-PCR revealed a similar pattern of expression as in the RNA-Seq experiment. These outcomes further support the reliability of the study and strongly suggest these genes as candidates for further investigation into their roles as target genes in C3G treatment. Furthermore, we propose that the upregulation of the six antioxidant genes in the C3G treatment group can be a major molecular mechanism providing antioxidant properties. Additionally, no antioxidant genes were found to be downregulated in the C3G-treated group. These results justify the current approach (Figure 3B), suggesting that the upregulation of antioxidant genes (S100a8, S100a9, Prdx2, Hp, Mpst, and Prxl2a) may represent a new avenue for further exploration of the antioxidant mechanisms of C3G in different diseases. Additionally, these antioxidant genes are associated with immune responses, especially inflammation. Most of these genes (S100a8, S100a9, Prdx2, Hp, and Mpst) were also associated with AD in earlier studies [74,75,76,77,78,79]. Furthermore, the upregulation of 28 genes in the immune system database and 33 in the innate gene DBs could have directly contributed to the acquired and innate immunity in the ADM mice+C3G group, respectively. The upregulation of immunity, especially innate immunity, might be one of the critical mechanisms that support the immune-regulation effects of C3G and the subsequent impact on AD and other diseases [38]. The upregulation of antioxidant and immunity genes was found to be a possible mechanism creating the antioxidant and immune-modulatory properties of C3G treatment in the current study. Further studies are needed to explore the precise role of upregulated and common DEGs to optimize the therapeutic potential of C3G. Although a similar response from C3G treatment can be expected in different tissues, it may or may not be the same. Therefore, further studies using other tissues are recommended according to the disease and/or the condition under investigation. In addition, C3G-treated wild-type mice can be included in future studies for comparison with the ADM mice+C3G group, which may be helpful to better understand the therapeutic effects of C3G and the subsequent impact on AD.

## 5. Conclusions

For the first time, a comparative transcriptome analysis of the spleen obtained from a C3G-treated mouse model of AD was conducted. Some DEGs important for immunity and antioxidant activity were found to be common in both comparisons, suggesting their pivotal roles in AD pathogenesis and the need for further in vitro and in vivo experiments. An upregulation of antioxidant and immuno-related genes was observed in the C3G-treated mice, which might be important antioxidant and immune modulation mechanisms in C3G therapy. These findings underscore the need to further explore these specific genes as therapeutic targets in C3G treatment for AD and other important diseases such as cancer.

## Figures and Tables

**Figure 1 antioxidants-10-01435-f001:**
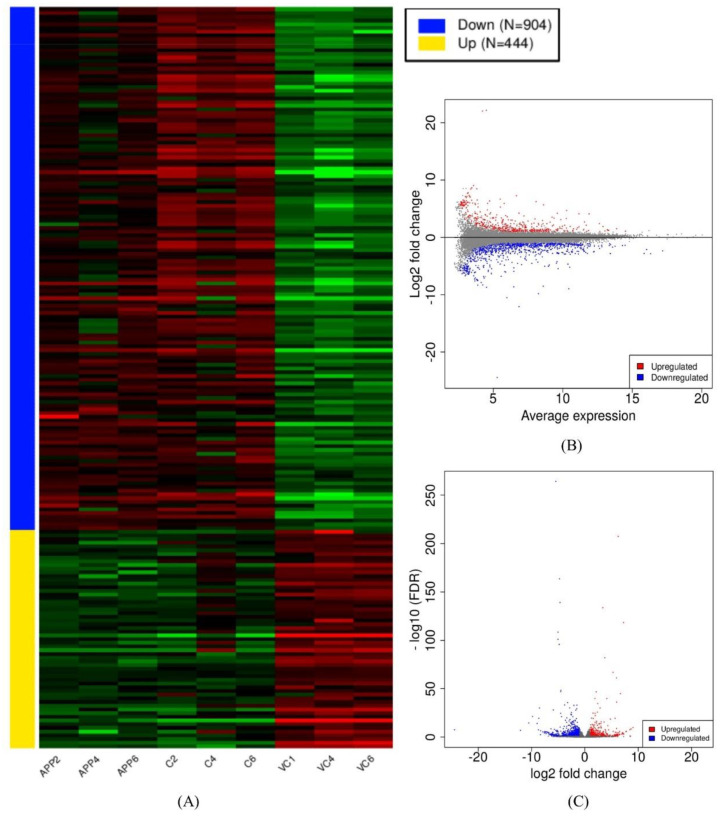
(**A**) Heatmap, (**B**) MA, and (**C**) volcano plots of differentially expressed genes (DEGs) in the comparison of Wt mice vs. ADM mice.

**Figure 2 antioxidants-10-01435-f002:**
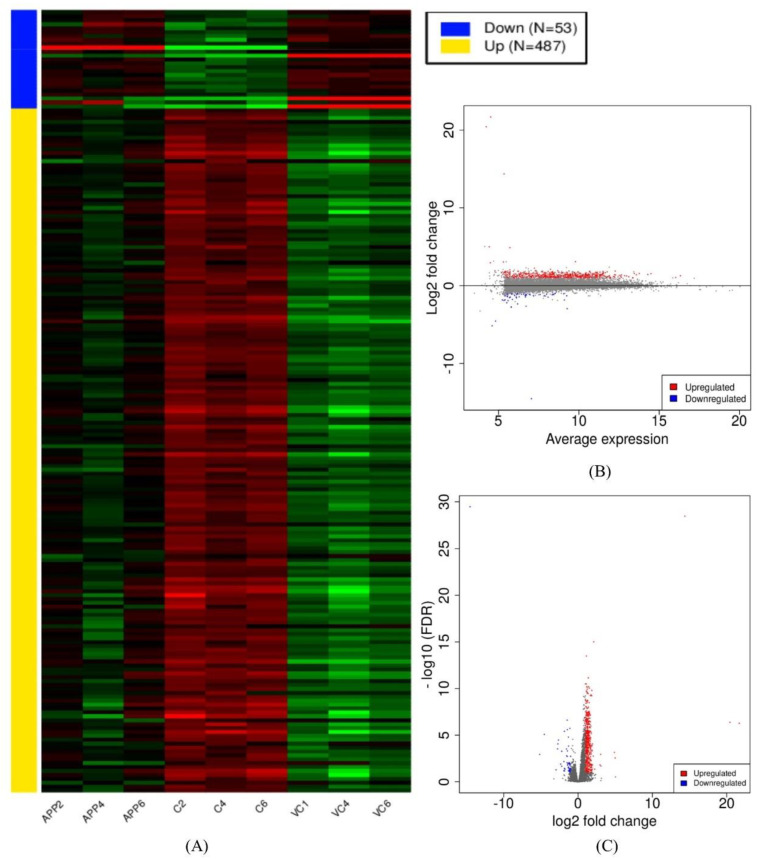
(**A**) Heatmap, (**B**) MA, and (**C**) volcano plots of differentially expressed genes (DEGs) in comparison of ADM mice+C3G vs. ADM mice.

**Figure 3 antioxidants-10-01435-f003:**
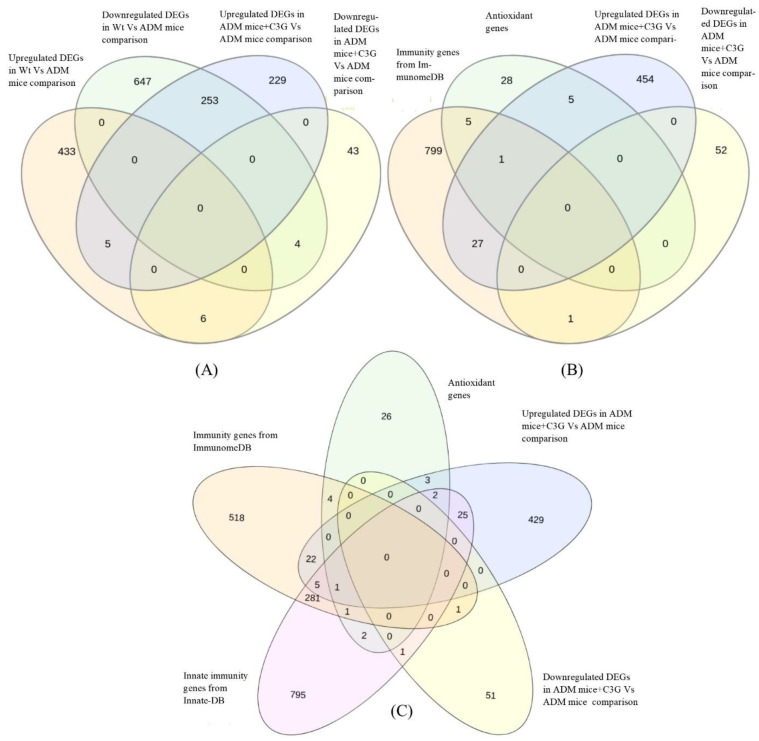
(**A**) Venn diagram of DEGs based on Wt mice vs. ADM mice and ADM mice+C3G vs. ADM mice comparisons. (**B**) Venn diagram of DEGs (Wt mice vs. ADM mice) overlapping with immunome database and genes associated with innate immunity from InnateDB. (**C**) Venn diagram between DEGs (ADM mice+C3G vs. ADM mice) with antioxidant, immunome database, and innate immune-related genes.

**Figure 4 antioxidants-10-01435-f004:**
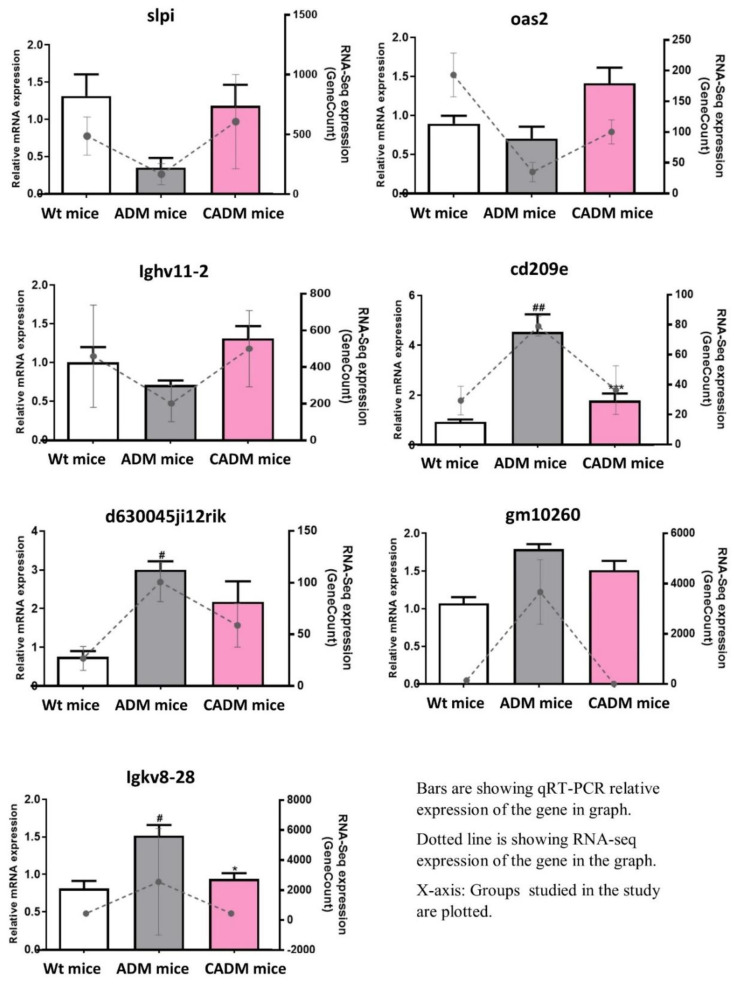
qRT-PCR expression of selected genes in all three groups plotted with RNA-Seq expression. # *p* < 0.05, ## *p* < 0.01 vs. wild-type mice (Wt mice); * *p* < 0.05 and *** *p* < 0.001 vs. ADM mice+C3G (CADM mice).

**Table 1 antioxidants-10-01435-t001:** Preprocessing and alignment results of all nine samples.

No.	Group	Sample ID	Number of Reads	Number of Good Reads (Percentage)	Overall Alignment Rate
1	Wt mice	VC1	21,622,102	20,743,123 (95.9%)	93.51%
2		VC4	23,804,636	22,983,122 (96.5%)	94.19%
3		VC6	25,529,616	24,586,395 (96.3%)	94.10%
4	ADM mice	APP2	20,475,388	19,700,219 (96.2%)	93.03%
5		APP4	20,315,569	19,541,861 (96.1%)	91.00%
6		APP6	24,571,206	23,772,655 (96.7%)	93.35%
7	ADM mice+C3G	C2	25,878,453	25,060,447 (96.8%)	91.31%
8		C4	25,609,508	24,844,294 (97.01%)	93.06%
9		C6	24,938,833	24,141,244 (96.8%)	92.61%
10	All samples		Total: 212,745,311 Average: 23,638,367.8	Total: 205,373,360	Mean: 92.906%

**Table 2 antioxidants-10-01435-t002:** Differentially expressed genes proposed to be important candidate in C3G -induced antioxidant and/or immune modulation activity.

Sr. No.	Gene Name	Differential Expression	Associated with Immune Function	Associated with Antioxidant Activity	Associated with AD
1	Slpi	Upregulated in both the comparisons	Yes	Yes	Yes
2	Oas2		Yes		
3	Ighv11-2		Yes		
4	Nnt		Yes	Yes	Yes
5	S100a8	Upregulated in comparison II (i.e., treatment group)	Yes	Yes	Yes
6	S100a9		Yes	Yes	Yes
7	Prdx2		Yes	Yes	Yes
8	Hp		Yes	Yes	Yes
9	Mpst			Yes	Yes
10	Prxl2a			Yes	

## Data Availability

RNA-Seq data generated and utilized in the study is submitted to publically accessible NCBI repository (BioProject: PRJNA749157).

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
