# Peer review of "Comparative Transcriptome Analysis of the Expression of Antioxidant and Immunity Genes in the Spleen of a Cyanidin 3-O-Glucoside-Treated Alzheimer’s Mouse Model"

_antioxidants, 2021, doi:10.3390/antiox10091435_

Round 1

Reviewer 1 Report

in the manuscript authored by Varun Jaiswal et al, authors explored with a transcriptomic approach how a cyanidin 3-O- 3 glucoside treatment modifies the expression of antioxidant and immunity genes in an AD mice model. 

the manuscript is well-structured and interesting, but here below I reported my comments and suggestions:

-please modify in the manuscript the use of acronyms as normal (NM), Alzheimer’s mouse model (AM), and C3G-treated AM mice (CAM). A suggestion could be mice, AD mice, AD mice+C3G.

-Immediately state in the abstract which AD model is used in the study.

-statistical analysis is missing in methods.

-Avoid using the acronym DEG for (Differentially expressed genes)

-please modify figure 5 since it is misleading. Delete blue lines since this study it is not a follow-up study. 

-How authors selected the dosage and AD mice model used in the study? add in the text

-C3G treatment improved cognitive performance in AD models?

Author Response

R1: Comments and Suggestions for Authors

in the manuscript authored by Varun Jaiswal et al., authors explored with a transcriptomic approach how a cyanidin 3-O- 3 glucoside treatment modifies the expression of antioxidant and immunity genes in the AD mice model.

the manuscript is well-structured and interesting, but here below I reported my comments and suggestions:

-please modify in the manuscript the use of acronyms as normal (NM), Alzheimer’s mouse model (AM), and C3G-treated AM mice (CAM). A suggestion could be mice, AD mice, AD mice+C3G.

Reply: Thank you for the suggestion. As per the suggestions from the reviewers for the modification of acronyms, in the updated manuscript, the following change has been considered (Wt mice, ADM mice, and ADM mice+C3G). As per comments of reviewer-2 and reviewer-3, wild-type, ‘Wt’ (Wt mice) has been added for C57BL/6J Jms mice, and model ‘M’ (ADM mice) has been added with AD mice. (Line no. 90-95).

Line no. 90-95; ‘Animals used in this study were: wild-type mice (C57BL/6J Jms) purchased from Hamamatsu-shi, Shizuoka, Japan, grouped as Wt–mice; AD model mice which were APPswe/PS1dE9 mice purchased from the Jackson laboratory designated as ADM mice. ….. and AMM treated with C3G grouped as ADM mice+C3G.

-Immediately state in the abstract which AD model is used in the study.

Reply: Thank you for the important suggestion. As per the reviewer’s suggestion, the AD model has been added in the abstract in the revised manuscript (Line no. 15 and 17).

Line no. 15; ‘Alzheimer’s disease (AD)’

Line no. 17; ‘Alzheimer’s mouse model (APPswe/PS1dE9 mice)’

-statistical analysis is missing in methods.

Reply: Thank you for the important suggestion. As per the reviewer’s suggestion, the standard methods have been used, which incorporate statistical analysis in the calculation, statistical analysis has been highlighted in the modified manuscript. (Line no. 141 and 180)

Line no. 141; ‘The default statistical cutoff parameters for identifying DEGs (FDR < 0.1 and minimum 2-fold change) were used in DESeq2.’

Line no. 180; ‘The default setting was utilized for the study, a statistical parameter such as p-value for analysis was calculated, and the results were extracted in tab-separated variables and PNG image formats.’

-Avoid using the acronym DEG for (Differentially expressed genes)

Reply: As per the reviewer’s suggestion, the acronym DEG was replaced with DEGs in the revised manuscript (Line no. 144, 255, 264, and 265).

Line no. 144; ‘Differentially expressed genes (DEGs)’

Line no. 255; ‘Identification of DEGs with antioxidant activity

Line no. 264; ‘Identification of DEGs with the immune-related function’

Line no. 265; ‘In comparing DEGs with ImmGS1’

-please modify figure 5 since it is misleading. Delete blue lines since this study it is not a follow-up study.

Reply: As per the reviewer’s suggestion, figure 5 has been changed in the revised manuscript (refer to figure 4).

-How authors selected the dosage and AD mice model used in the study? add in the text

Reply: Thank you for your comment. The 50% lethal dose (LD50) of anthocyanin–lauric acid derivatives (ALDs, the main component of ALDs is C3G) in mice was more than 10 g/kg (Yang et al. 2020). And the subacute toxicity test results demonstrated that less than 600 mg/kg of ALDs intake did not affect mortality. In the Alzheimer’s mouse model, specially APPswe/PS1dE9 mice, after 6 to 7 months of age, mice develop beta-amyloid deposits in the brain. Deficits in cognitive functions were seen in 12-13 months with the Morris Water Maze test (Malm et al., 2011) in these APPswe/PS1dE9 mice. Beta-amyloid plaques grew faster in 6-month-old compared with 10-month-old mice (Yan et al., 2009). Before 10 month-old mice, we wanted to know how much C3G made improved immune responses along with oral administration for five to nine months. After assessing the Y-maze test (unpublished data), we finished the animal experiment (Lau et al., 2008 and Snellman et al.,2019). The C3G is not a drug and is the constituent present in normal foods (fruits, vegetables, and grains); hence, its long period of dosage/treatment is possible.

In earlier research work, the different dosages of C3G (5 mg/kg/day of C3G for more than 8 weeks (Song et al., 2016), 50 mg/kg/day of C3G for 4 weeks or 8 weeks (Wang et al., 2012, Jia et al., 2020) and 6 mg/mouse/day (300 mg/kg/day) of C3G for 22 weeks (Amararathna et al., 2020)) has been used, which was found to be effective. So, our experiment selected the dose of C3G (30 mg/kg/day) for 38 weeks. Details with references have been added in the revised manuscript. (Line no. 98-101)

Line no. 98-101; ‘The C3G dose orally administered to the ADM mice+C3G group was 30 mg/kg/day for 38 weeks. The mice were provided ad libitum access to food and water during this study. The dose and study period were decided according to earlier experiments which investigated the effect of C3G in similar studies [1, 9, 27, 43].’

Yang et al., Food Funct. 2020 Dec 1;11(12):10954-10967.

Malm et al., Int J Alzheimers Dis. 2011; 2011: 517160.

Yan et al., J Neurosci. 2009 Aug 26; 29(34): 10706–10714.

Lau et al., Neuroimage. 2008 Aug 1;42(1):19-27.

Snellman et al., Sci Rep. 2019 Apr 5;9(1):5700.

Wang et al., Circ Res. 2012 Sep 28;111(8):967-81.

Jia et al., Commun Biol. 2020 Sep 18;3(1):514.

Amararathna et al., Molecules. 2020 Aug 22;25(17):3823.

-C3G treatment improved cognitive performance in AD models?

Reply: Yes, as per earlier literature (Song et al., 2016) and in current work, the C3G treatment improved cognitive performance in AD models. (Line no. 304 and 305)

Line no. 304-305; ‘In an earlier study, C3G also alleviated the cognitive impairment in the same Alzheimer's mouse model.’

But now, we are preparing a second paper about the different study designs of concentration of amyloid-beta (Aβ) plaque and microglia cell distribution after treatment C3G.

Reviewer 2 Report

In the manuscript by Jaiswal et al the authors investigated the antioxidant and anti-inflammatory potential of cyanidin 3-O-glucoside (C3G) through a whole transcriptome analysis. The study aims to explore the differential expression of antioxidant and immune genes in a mouse model of AD.

Major comments:

One of my main concerns regarding this study is why did the authors choose to perform the study using spleen and not brain tissue? Since the study uses an Alzheimer´s disease model the obvious choice was to include brain tissue samples, even if the spleen was also analysed, as being relevant for immune response. How do the authors envision that the analysis of the spleen can be translatable and relevant for AD?

Line 82: The mice were injected every day for 38 weeks??? This seems excessive. Or this was not the route of administration? The administration route should be clearly stated. Did any other protocols were tried? Why subject the animals to such a long treatment? A clear justification for this should be presented.

Attention!!! Throughout the whole text the authors refer “mice with AD” or “Alzheimer´s mice”. This must be altered. It is scientifically INCORRECT to say “AD mice” since the mice do not develop Alzheimer´s disease. It is a MODEL of AD, and not the disease itself (e.g. line 61; line 66).

Instead of presenting so many “pre-processing results” (Figure1), that could be only in supplementary material, the authors could have included tables and schemes that helped the reader to clearly know which DEGs were identified and which genes were validated. This would improve readability.

In the Discussion the authors state “to identify the genes and molecular mechanisms…” (Line 266) and “…was carried out to explore the genes and the mechanism of antioxidant and immune modulation” (Line 269). However, no molecular mechanisms are investigated in the present work. There is merely gene identification. Molecular pathways or mechanisms are only briefly suggested/mentioned in the discussion but are not obvious from the results and were not highlighted in the figures presented.

The language used throughout the whole manuscript has flaws that should be corrected before it can be considered for publication. Some examples, among others in the manuscript, can be found in the comments below.

Minor comments

Lines 32-35: please reformulate this sentence as in the beginning different diseases/conditions are being enumerated and in the end of the sentence are “antioxidant and immunomodulatory activities”. Sentence is not well structured.

Lines 39-41 and 45-46: again, there are flaws in language and sentences are not well structured and/or do not use appropriate scientific terms.

Lines 46-47: this sentence (“Natural compounds are preferred as therapeutics due to their pharmacokinetic profile compared with synthetic compounds”) is an extrapolation. I am sure it depends on the natural vs synthetic compounds that are being analysed. Furthermore, the sentence is not supported by scientific studies and no references were added. My suggestion is to remove this sentence.

Line 55: a reference to the studies analysing other types of cell-death, rather than the Aβ-induced, is missing.

Line 61: “mice with AD” do not exist. It´s not accurate to say this since mice do not develop Alzheimer´s disease. The authors meant a mouse model of AD. This should be altered.

Line 72: “The current study methodology involves laboratory experiments (animal studies, transcriptome sequencing, and qRT-PCR) and data analysis (RNA-Seq data and other computational analysis) …” This is an obvious statement. In general, all scientific papers in this area have a methodological component and a subsequent data analysis.

Line 77: “normal mice” is incorrect, should be “wild type mice”

Author Response

R2: Comments and Suggestions for Authors

In the manuscript by Jaiswal et al., the authors investigated the antioxidant and anti-inflammatory potential of cyanidin 3-O-glucoside (C3G) through whole transcriptome analysis. The study aims to explore the differential expression of antioxidant and immune genes in a mouse model of AD.

Major comments:

One of my main concerns regarding this study is why did the authors choose to perform the study using spleen and not brain tissue? Since the study uses an Alzheimer´s disease model, the obvious choice was to include brain tissue samples, even if the spleen was also analysed, as being relevant for immune response. How do the authors envision that the analysis of the spleen can be translatable and relevant for AD?

Reply: Thank you for your valuable comment. Brain tissues are majorly involved in the nervous system-related function, which may overshadow the immunity and antioxidant gene expression. Before our study, Song et al. (2016) showed the protective effects of C3G on Aβ25–35 induced cell apoptosis in SH-SY5Y cells and improved cognitive impairment in the APPswe/PS1dE9 mouse model of AD. After that, we focused on the antioxidant and immunity genes expression in the C3G treatment to know the critical target genes for the antioxidant and immune modulation of the spleen in the APPswe/PS1dE9 mouse model of AD after administration of C3G. Spleens displayed accumulation of amyloid-β1–42 (Aβ1-42) and high expression of Treg cell markers FoxP3 and GITR, in parallel with the increased levels of inflammatory markers expression. And neuroinflammation in the brain might result in a general overshooting of the immune response in the whole body (Di Benedetto et al., 2019). In general, a similar outcome can be expected in other tissues, but further research may be required to confirm.

Line no. 358-361; ‘Although a similar response from C3G treatment can be expected in different tissue, it may or may not be the same. Therefore, further studies with other tissues are recommended according to the disease or the condition under investigation.

Di Benedetto et al. 2019. J Neuroinflammation 16, 166.

Line 82: The mice were injected every day for 38 weeks??? This seems excessive. Or was this not the route of administration? The administration route should be clearly stated. Did any other protocols were tried? Why subject the animals to such a long treatment? A clear justification for this should be presented.

Reply: Thank you for your comment. The 50% lethal dose (LD50) of anthocyanin–lauric acid derivatives (ALDs, the main component of ALDs is C3G) in mice was more than 10 g/kg (Yang et al. 2020). And the subacute toxicity test results demonstrated that less than 600 mg/kg of ALDs intake did not affect mortality. In the Alzheimer’s mouse model, specially APPswe/PS1dE9 mice, after 6 to 7 months of age, mice develop beta-amyloid deposits in the brain. Deficits in cognitive functions were seen in 12-13 months with the Morris Water Maze test (Malm et al., 2011) in these APPswe/PS1dE9 mice. Beta-amyloid plaques grew faster in 6-month-old compared with 10-month-old mice (Yan et al., 2009). Before 10 month-old mice, we wanted to know how much C3G made improved immune responses along with oral administration for five to nine months. After assessing the Y-maze test (unpublished data), we finished the animal experiment (Lau et al., 2008 and Snellman et al.,2019). The C3G is not a drug and is the constituent present in normal foods (fruits, vegetables, and grains); hence, its long period of dosage/treatment is possible.

In earlier research work, the different dosages of C3G (5 mg/kg/day of C3G for more than 8 weeks (Song et al., 2016), 50 mg/kg/day of C3G for 4 weeks or 8 weeks (Wang et al., 2012, Jia et al., 2020) and 6 mg/mouse/day (300 mg/kg/day) of C3G for 22 weeks (Amararathna et al., 2020)) have been used, which was found to be effective. So, our experiment selected the dose of C3G (30 mg/kg/day) for 38 weeks. Details with references have been added in the revised manuscript. (Line no. 98-101)

Line no. 98~101; ‘The C3G dose orally administered to the CADM mice group was 30 mg/kg/day for 38 weeks. The mice were provided ad libitum access to food and water during this study. The dose and study period were decided according to earlier experiments which investigated the effect of C3G in similar studies [1, 9, 27, 43].’

Yang et al., Food Funct. 2020 Dec 1;11(12):10954-10967.

Malm et al., Int J Alzheimers Dis. 2011; 2011: 517160.

Yan et al., J Neurosci. 2009 Aug 26; 29(34): 10706–10714.

Lau et al., Neuroimage. 2008 Aug 1;42(1):19-27.

Snellman et al., Sci Rep. 2019 Apr 5;9(1):5700.

Wang et al., Circ Res. 2012 Sep 28;111(8):967-81.

Jia et al., Commun Biol. 2020 Sep 18;3(1):514.

Amararathna et al., Molecules. 2020 Aug 22;25(17):3823.

Attention!!! Throughout the whole text, the authors refer to “mice with AD” or “Alzheimer´s mice.” This must be altered. It is scientifically INCORRECT to say “AD mice” since the mice do not develop Alzheimer´s disease. It is a MODEL of AD and not the disease itself (e.g., line 61; line 66).

Reply: We are so sorry about that. As per the reviewer’s suggestion, the correction has been made in the revised manuscript. (Line no. 72, 77, 94, and 370)

Line no. 72; ‘mouse model of AD’

Line no. 77; ‘mouse model of AD’

Line no. 94; ‘mouse model of AD’

Line no. 370; ‘mouse model of AD’

Instead of presenting so many “pre-processing results” (Figure 1) that could be only in supplementary material, the authors could have included tables and schemes that helped the reader to clearly know which DEGs were identified and which genes were validated. This would improve readability.

Reply: Thank you for your comment. As per the reviewer’s suggestion, figure 1 has been transferred to supplementary materials in the updated submission, and Table 2 has been added in the result section of the revised manuscript, which highlights the important upregulated DEGs (Line no. 249).

Line no. 249; ‘Table 2. Differentially expressed genes proposed as an important candidate in C3G induced anti-oxidant and/or immune modulation activity.’

In the Discussion, the authors state “to identify the genes and molecular mechanisms…” (Line 266) and “…was carried out to explore the genes and the mechanism of antioxidant and immune modulation” (Line 269). However, no molecular mechanisms are investigated in the present work. There is merely gene identification. Molecular pathways or mechanisms are only briefly suggested/mentioned in the discussion but are not obvious from the results and were not highlighted in the figures presented.

Reply: Thank you for your comment. As per the reviewer’s suggestion, the sentence has been reframed in the revised manuscript (Line no. 298 and 301). Before our study, Song et al. (2016) showed the protective effects of C3G on Aβ25–35-induced cell apoptosis in SH-SY5Y cells and improved cognitive impairment in the APPswe/PS1dE9 mouse model of AD. Spleens displayed accumulation of amyloid-β1–42 (Aβ1-42) and high expression of Treg cell markers FoxP3 and GITR, in parallel with the increased levels of inflammatory markers expression. And neuroinflammation in the brain might result in a general overshooting of the immune response in the whole body (Di Benedetto et al., 2019). So, we wanted to focus on the antioxidant and immunomodulatory activities of C3G the spleen in the APPswe/PS1dE9 mouse model of AD.

Line no. 298; ‘Conducted comparative whole transcriptome analysis of C3G-treated AD mice to identify the genes and subsequently infer the potential molecular mechanisms underlying the antioxidant and immunomodulatory properties,..’.

Line no. 301; Further analysis considering the importance of these properties in various diseases was carried out to explore the genes and the potential mechanism of antioxidant and immune modulation to develop C3G as a possible therapeutic not only in AD but also in other conditions.

In addition, we added a previous study about C3G in the same mouse model of AD in lines 304~305.

Line no. 304~305; ‘In an earlier study, C3G also alleviated the cognitive impairment in 297 the same Alzheimer's mouse model [26].’

The language used throughout the whole manuscript has flaws that should be corrected before it can be considered for publication. Some examples, among others in the manuscript, can be found in the comments below.

Reply: Thank you for your comment. We already had an editing service, and we’ll get a professional editing service again before publication. 

Minor comments

Lines 32-35: please reformulate this sentence as in the beginning, different diseases/conditions are being enumerated and at the end of the sentence are “antioxidant and immunomodulatory activities.” A sentence is not well structured.

Reply: Thank you for your kind advice. As per the reviewer’s suggestion, the mentioned sentence has been reframed and divided into two sentences. The second sentence has been reframed and merged with the next sentence in the revised manuscript (Line no. 33~39).

Line no. 33~39; C3G was found to have protective and therapeutic potential in diabetes [1-3], obesity [4], cardiovascular disease [2,5], neurological diseases [6], asthma [7], atherosclerosis [8], Alzheimer’s disease (AD) [6] and, different types of cancer [9-13]. These multiple pharmacological properties of C3G may be primarily attributed to the antioxidant and immunomodulatory effects of C3G, which have been reported in different studies [10,14-16].

Lines 39-41 and 45-46: again, there are flaws in the language, and sentences are not well structured and/or do not use appropriate scientific terms.

Reply: Thank you for your kind advice. The sentences have been reframed and deleted accordingly in the revised manuscript (Line no. 43~46)

Line no. 43~46; ‘Therefore, the current study was designed to investigate the effect of C3G on the whole transcriptome expression of the spleen to discover important immune-modulating and antioxidant genes using wild-type and AD mouse model.

Lines 46-47: this sentence (“Natural compounds are preferred as therapeutics due to their pharmacokinetic profile compared with synthetic compounds”) is an extrapolation. I am sure it depends on the natural vs. synthetic compounds that are being analysed. Furthermore, the sentence is not supported by scientific studies, and no references were added. My suggestion is to remove this sentence.

Reply: Thank you for your valuable comment. The mentioned sentence was removed in the revised manuscript (Line no. 57).

Line 55: a reference to the studies analysing other types of cell death, rather than the Aβ-induced, is missing.

Reply: Thank you for your comment. As per the reviewer’s concern, we have added references for cell death (Line no. 66).

Line 61: “mice with AD” do not exist. It´s not accurate to say this since mice do not develop Alzheimer´s disease. The authors meant a mouse model of AD. This should be altered.

Reply: We are so sorry about that. As per the reviewer’s suggestion, the corrections have been made in the revised manuscript (Line no. 72, 77, 94, and 370).

Line no. 72; ‘mouse model of AD’

Line no. 77; ‘mouse model of AD’

Line no. 94; ‘mouse model of AD’

Line no. 370; ‘mouse model of AD’

Line 72: “The current study methodology involves laboratory experiments (animal studies, transcriptome sequencing, and qRT-PCR) and data analysis (RNA-Seq data and other computational analysis) …” This is an obvious statement. In general, all scientific papers in this area have a methodological component and a subsequent data analysis.

Reply: We are very sorry for the mistake. As per the reviewer’s suggestion, these statements have been removed from the revised manuscript (Line no 84~87).

Line 77: “normal mice” is incorrect, it should be “wild type mice”

Reply: Thank you for your kind advice. Corrections have been made in the revised manuscript (Line no. 91).

Line no. 91; ‘wild-type mice (C57BL/6J Jms)’

Reviewer 3 Report

Introduction

46-47 Lines: «Natural compounds are preferred as therapeutics due to their pharmacokinetic profile compared with synthetic compounds.» This sentence is nonsense. Please, remove it.

42-44 Lines: «AD is the sixth leading cause of death in the United States (US) [14], with an estimated more than 6 million individuals diagnosed with AD in the age group of 65 years or above in 2021 [15].» Being authors from Korea, why don’t they include figures about Korea? It seems the logical thing to do.

Materials and Methods

It is not appropriate to use the adjective normal (mice) for the Wild-type mice bred on the same genetic background C57BL/6J. Authors should use wild-type mice.

Origin and source of the C3G should be indicated in material and methods.

The C3G dose administered to the CAM group was 30 mg/kg/day for 38 weeks. Authors should explain the rationale of using such treatment schedule and dose of C3G, and provide a reference.

Design of the study

It is true that standard pipelines were used in this study for the identification of gene expression and differential gene expression analysis.

Authors should consider the introduction of a wild-type group of mice receiving the C3G treatment. The treatment intervention in the transgenic mice should be compared with that of the wild-type mice in order to better understand (for instance) the therapeutically meaning of the immune regulation effects of C3G and the subsequent impact on AD and/or other diseases.

Author Response

R3: Comments and Suggestions for Authors

Introduction

46-47 Lines: «Natural compounds are preferred as therapeutics due to their pharmacokinetic profile compared with synthetic compounds.» This sentence is nonsense. Please, remove it.

Reply: Thank you for your valuable comment. As per the reviewer’s suggestion, the sentence has been removed (Line no. 55~58).

42-44 Lines: «AD is the sixth leading cause of death in the United States (US) [14], with an estimated more than 6 million individuals diagnosed with AD in the age group of 65 years or above in 2021 [15].» Being authors from Korea, why don’t they include figures about Korea? It seems the logical thing to do.

Reply: Thank you for your comment. The suggestion has been incorporated in the revised manuscript (Line no.51~52).

Line no. 51~52; ‘In Korea, AD was reported to be the ninth leading cause of death which was expected to increase. AD is the sixth leading cause of death in the United States (US)...’

Materials and Methods

It is not appropriate to use the adjective normal (mice) for the Wild-type mice bred on the same genetic background C57BL/6J. Authors should use wild-type mice.

Reply: We are very sorry for the mistake. As per the reviewer’s concern, the suggestion has been implemented in the updated manuscript (Line no. 91).

Line no. 91; ‘wild-type mice (C57BL/6J Jms) purchased from Hamamatsu-shi, Shizuoka, …’

The origin and source of the C3G should be indicated in material and methods.

Reply: Thank you for your comment. The C3G was purchased from the ChemFaces, and the detailed sources were added to the revised manuscript (Line no. 89~90). 

Line no. 89~90; ‘C3G (Cyanidin-3-O-glucoside (C21H21O11, Cat. No. CFN99740) used in the study was purchased from ChemFaces (Wuhan, Hubei, China).’

The C3G dose administered to the CAM group was 30 mg/kg/day for 38 weeks. The authors should explain the rationale of using such a treatment schedule and dose of C3G and provide a reference.

Reply: Thank you for your comment. The 50% lethal dose (LD50) of anthocyanin–lauric acid derivatives (ALDs, the main component of ALDs is C3G) in mice was more than 10 g/kg (Yang et al. 2020). And the subacute toxicity test results demonstrated that less than 600 mg/kg of ALDs intake did not affect mortality. In the Alzheimer’s mouse model, specially APPswe/PS1dE9 mice, after 6 to 7 months of age, mice develop beta-amyloid deposits in the brain. Deficits in cognitive functions were seen in 12-13 months with the Morris Water Maze test (Malm et al., 2011) in these APPswe/PS1dE9 mice. Beta-amyloid plaques grew faster in 6-month-old compared with 10-month-old mice (Yan et al., 2009). Before 10 month-old mice, we wanted to know how much C3G made improved immune responses along with oral administration for five to nine months. After assessing the Y-maze test (unpublished data), we finished the animal experiment (Lau et al., 2008 and Snellman et al.,2019). The C3G is not a drug and is the constituent present in normal foods (fruits, vegetables, and grains); hence, its long period of dosage/treatment is possible.

In earlier research work, the different dosages of C3G (5 mg/kg/day of C3G for more than 8 weeks (Song et al., 2016), 50 mg/kg/day of C3G for 4 weeks or 8 weeks (Wang et al., 2012, Jia et al., 2020) and 6 mg/mouse/day (300 mg/kg/day) of C3G for 22 weeks (Amararathna et al., 2020)) has been used, which was found to be effective. So, our experiment selected the dose of C3G (30 mg/kg/day) for 38 weeks. Details with references have been added in the revised manuscript. (Line no. 98~101)

Line no. 98~101; ‘The C3G dose orally administered to the CADM mice group was 30 mg/kg/day for 38 weeks. The mice were provided ad libitum access to food and water during this study. The dose and study period were decided according to earlier experiments which investigated the effect of C3G in similar studies [1, 9, 27, 43].’

Yang et al., Food Funct. 2020 Dec 1;11(12):10954-10967.

Malm et al., Int J Alzheimers Dis. 2011; 2011: 517160.

Yan et al., J Neurosci. 2009 Aug 26; 29(34): 10706–10714.

Lau et al., Neuroimage. 2008 Aug 1;42(1):19-27.

Snellman et al., Sci Rep. 2019 Apr 5;9(1):5700.

Wang et al., Circ Res. 2012 Sep 28;111(8):967-81.

Jia et al., Commun Biol. 2020 Sep 18;3(1):514.

Amararathna et al., Molecules. 2020 Aug 22;25(17):3823.

Design of the study

It is true that standard pipelines were used in this study for the identification of gene expression and differential gene expression analysis.

The authors should consider the introduction of a wild-type group of mice receiving the C3G treatment. The treatment intervention in the transgenic mice should be compared with that of the wild-type mice in order to better understand (for instance) the therapeutically meaning of the immune regulation effects of C3G and the subsequent impact on AD and/or other diseases.

Reply: Thank you for your valuable comment. At that time, we did not have enough grants to conduct experiments. We totally agree with the reviewer’s opinion. We would like to incorporate a wild-type group of mice aspects in our further study.

Round 2

Reviewer 1 Report

Authors provided most of the corrections requested by the reviewer thus now the revised version of the manuscript can be considered for a possible publication in Antioxidants

Author Response

Reviewer 1

Open Review

English language and style

( ) Extensive editing of English language and style required
( ) Moderate English changes required
(x) English language and style are fine/minor spell check required
( ) I don't feel qualified to judge about the English language and style

Authors provided most of the corrections requested by the reviewer thus now the revised version of the manuscript can be considered for a possible publication in Antioxidants

We appreciate you for taking the time to review our study. We had spell check, and English grammar in the manuscript has been revised by a professional.

Reviewer 2 Report

In the manuscript by Jaiswal et al the authors aim to explore the differential expression of antioxidant and immune genes in response to cyanidin 3-O-glucoside (C3G) in a mouse model of AD using a whole transcriptome analysis.

In the revised manuscript the authors made an effort to include the reviewers’ suggestions and by doing that they have improved the quality of the manuscript which, to my opinion, can now be considered for publication should the authors include some suggestions that follow:

In section 2.1 instead of “Chemical and Animals” would be better something like “Materials and Animal Model” or just “Animal model used in the study” since the chemical used is part of the protocol used in the model… not necessary to state that in the section title.

In the Discussion the initial sentence (lines 302-305) is not “clear”, should be altered to something like: “The comparative whole transcriptome analysis of C3G-treated AD mice conducted in the present study allowed to identify the genes and subsequently infer the possible molecular mechanisms underlying the antioxidant and immunomodulatory properties of C3G previously observed in a number of studies”

Also in the Discussion, the authors have appropriately replied to my comments regarding the molecular mechanisms. The content of their reply is satisfactory but should also be included in the discussion, so the information is made available for the readers and not only for the reviewers. The authors have included a sentence (lines 308-309) regarding cognitive impairment, but more importantly the statement “And neuroinflammation in the brain might result in a general overshooting of the immune response in the whole body (Di Benedetto et al., 2019). So, we wanted to focus on the antioxidant and immunomodulatory activities of C3G the spleen in the APPswe/PS1dE9 mouse model of AD” that is part of their reply should also be included in the discussion. Moreover, the addition of this sentence to the manuscript will also help to partially address one of my main concerns regarding this study, that I have referred in the first revision report, and that is why did the authors choose to perform the study using spleen and not brain tissue. This issue was not fully addressed by the authors and their reply “Brain tissues are majorly involved in the nervous system-related function, which may overshadow the immunity and antioxidant gene expression.” is not a convincing explanation. I admit the authors which to explore the genes identified later in other studies, but then they can state that in the manuscript.

Regarding by comment to the route and time of administration of the compound, my concern was not so much the long period but rather if that corresponded to an injection to the animals every day, that would of course be a potential problem for the animal well-being. The authors have clarified that the route of administration is oral and that addresses my question. The route and dose of administration of C3G is now specified in lines 98-101 of the manuscript. But please state whether “orally” means “food intake” or “oral gavage”. This part of the manuscript has now improved and is clearer.

Author Response

Reviewer 2

Open Review

English language and style

( ) Extensive editing of English language and style required
( ) Moderate English changes required
(x) English language and style are fine/minor spell check required
( ) I don't feel qualified to judge about the English language and style

In the manuscript by Jaiswal et al the authors aim to explore the differential expression of antioxidant and immune genes in response to cyanidin 3-O-glucoside (C3G) in a mouse model of AD using a whole transcriptome analysis.

We appreciate you for taking the time to review our study. We had spell check, and English grammar in the manuscript has been revised by a professional.

In the revised manuscript the authors made an effort to include the reviewers’ suggestions and by doing that they have improved the quality of the manuscript which, to my opinion, can now be considered for publication should the authors include some suggestions that follow:

In section 2.1 instead of “Chemical and Animals” would be better something like “Materials and Animal Model” or just “Animal model used in the study” since the chemical used is part of the protocol used in the model… not necessary to state that in the section title.

à Thank you for your suggestion. We had changed ‘Chemical and Animals’ to ‘Materials and Animal Model’

Line no. 72; 2.1. Materials and Animal Model

In the Discussion the initial sentence (lines 302-305) is not “clear”, should be altered to something like: “The comparative whole transcriptome analysis of C3G-treated AD mice conducted in the present study allowed to identify the genes and subsequently infer the possible molecular mechanisms underlying the antioxidant and immunomodulatory properties of C3G previously observed in a number of studies”

à Thank you for your suggestion. The proposed sentence has been modified accordingly. (Line no. 268~271).

Line no. 268~271; “Comparative whole transcriptome analysis of C3G-treated ADM mice was conducted to identify genes and subsequently infer possible molecular mechanisms underlying the antioxidant and immunomodulatory properties of C3G observed in a number of studies [7,10,14,15,20,27].”

Also in the Discussion, the authors have appropriately replied to my comments regarding the molecular mechanisms. The content of their reply is satisfactory but should also be included in the discussion, so the information is made available for the readers and not only for the reviewers. The authors have included a sentence (lines 308-309) regarding cognitive impairment, but more importantly the statement “And neuroinflammation in the brain might result in a general overshooting of the immune response in the whole body (Di Benedetto et al., 2019). So, we wanted to focus on the antioxidant and immunomodulatory activities of C3G the spleen in the APPswe/PS1dE9 mouse model of AD” that is part of their reply should also be included in the discussion. Moreover, the addition of this sentence to the manuscript will also help to partially address one of my main concerns regarding this study, that I have referred in the first revision report, and that is why did the authors choose to perform the study using spleen and not brain tissue. This issue was not fully addressed by the authors and their reply “Brain tissues are majorly involved in the nervous system-related function, which may overshadow the immunity and antioxidant gene expression.” is not a convincing explanation. I admit the authors which to explore the genes identified later in other studies, but then they can state that in the manuscript.

à Thank you for your suggestion. Its relevance has not yet been well established. We revised the sentence in the discussion with your suggestion.

Line no. 275-278: The neuroinflammation in the brain might be associated with the immune response in the whole body [62]. So we wanted to focus on the antioxidant and immunomodulatory activities of C3G in the spleen of the APPswe/PS1dE9 mouse model of AD [63].

Regarding by comment to the route and time of administration of the compound, my concern was not so much the long period but rather if that corresponded to an injection to the animals every day, that would of course be a potential problem for the animal well-being. The authors have clarified that the route of administration is oral and that addresses my question. The route and dose of administration of C3G is now specified in lines 98-101 of the manuscript. But please state whether “orally” means “food intake” or “oral gavage”. This part of the manuscript has now improved and is clearer.

à Thank you for your comment. We had added line no. 80~81 as your comment.

Line no. 80~81; The C3G dose by oral gavage for the ADM mice+C3G group was 30 mg/kg/day for 38 weeks.

Reviewer 3 Report

Authors have replied appropriately every question except for one issue concerning the study design. Authors should consider the introduction of a wild-type group of mice receiving the C3G treatment.

Author Response

Reviewer 3

Open Review

English language and style

( ) Extensive editing of English language and style required
( ) Moderate English changes required
( ) English language and style are fine/minor spell check required
(x) I don't feel qualified to judge about the English language and style

We appreciate you for taking the time to review our study. We had spell check, and English grammar in the manuscript has been revised by a professional.

The authors have replied appropriately to every question except for one issue concerning the study design. The authors should consider the introduction of a wild-type group of mice receiving the C3G treatment.

à Thank you very much for your detailed comments. In our study, wild-type mice were used as the negative control for Alzheimer’s model mice as in other studies (Kim et al., 2021, Stoye et al., 2020). We just focused on the antioxidant effects of C3G in Alzheimer’s model mice.

Kim et al., Nat Commun. 2021 Apr 12;12(1):2185.

Stoye et al., FASEB J. 2020 Sep;34(9):11883-11899.

We will keep in mind your comment in our further study. Thank you again for understanding our present situation.  
